statistical physics/computer modelling and simulation/complexity

agent-based modelling, reinforcement learning, double auction markets, large deviations, effective single agent dynamics, Fokker–Planck equation

**Author for correspondence:**
Aleksandra Alorić
e-mail: aleksandra.aloric@gmail.com

# Fragmentation in trader preferences among multiple markets: market coexistence versus single market dominance

Robin Nicole[1], Aleksandra Alorić[2] and Peter Sollich[1,3]

[1]Department of Mathematics, King's College London, Strand, London WC2R 2LS, UK
[2]Scientific Computing Laboratory, Center for the Study of Complex Systems, Institute of Physics Belgrade, University of Belgrade, Pregrevica, 118, 11080 Belgrade, Serbia
[3]Institut für Theoretische Physik, Georg-August-Universität Göttingen, Friedrich-Hund-Platz 1, 37077 Göttingen, Germany

(iD) AA, 0000-0002-7278-599X

Technological advancement has led to an increase in the number and type of trading venues and a diversification of goods traded. These changes have re-emphasized the importance of understanding the effects of market competition: does proliferation of trading venues and increased competition lead to dominance of a single market or coexistence of multiple markets? In this paper, we address these questions in a stylized model of zero-intelligence traders who make repeated decisions at which of three available markets to trade. We analyse the model numerically and analytically and find that the traders' decision parameters—memory length and how strongly decisions are based on past success—make the key difference between consolidated and fragmented steady states of the population of traders. All three markets coexist with equal shares of traders only when either learning is too weak and traders choose randomly, or when markets are identical. In the latter case, the population of traders fragments across the markets. With different markets, we note that market dominance is the more typical scenario. Overall we show that, contrary to previous research emphasizing the role of traders' heterogeneity, market coexistence can emerge simply as a consequence of co-adaptation of an initially homogeneous population of traders.

# 1. Introduction

The possible risks and benefits of market competition have been the subject of a long-standing debate, which is often expressed as 'market consolidation versus market fragmentation' [1,2]. When the New York Stock Exchange had by far the strongest influence on price formation, the financial trading system was much closer to a consolidated state (Hasbrouck's [3]), but more recently technological progress has created a variety of trading venues and led to ever-increasing market fragmentation. Particularly interesting in this regard are the so-called *dark pools*. These trading venues have gained a certain notoriety from their lack of transparency and the possibility to trade large volumes without large price impacts, and they frequently offer a greater variety of market mechanisms compared to the conventional exchanges. Shorter & Miller [4] noted that in only five years (from 2008 to 2013) the US market share traded in dark pools increased from 4% to 15%, signalling a distinct increase in market fragmentation. Gomber *et al.* [1] suggest that the main driver of market fragmentation is the heterogeneity of traders' needs, which will be more easily satisfied by a variety of different markets rather than a single trading venue. In this paper, we show that even when identical markets compete, economic agents can develop loyalties to specific markets, thus effectively fragmenting trading. Conversely, we find in the case of competition of markets that are biased towards different classes within the population of traders, single market dominance is the typical outcome.

To tackle this question of market coexistence versus single market dominance, we build on previous work [5–8] where we introduced and analysed a system consisting of double auction markets and a large number of traders choosing between them. What we showed in this setting is that for a range of parameters describing the markets and agents, the agents split into groups with a strong loyalty towards one of the markets, often giving an overall market coexistence with an equal share of traders at both markets. When the agents have a long memory to previous trading outcomes, other steady states with single market dominance also exist and are in fact stable, whereas the system state with markets splitting trades roughly equally between them is only metastable [6,8]. While these initial studies focused on settings with two markets for simplicity, traders do in general have a choice between multiple markets (e.g. [1]) and this feature was also present in the CAT game [9] that originally motivated our research into market-trader co-fragmentation. We therefore extend the double auction market model from two to three markets in this paper, and use the results to formulate conjectures for the expected behaviour in cases where more than three markets compete.

There is a large body of work that uses the *JCAT* library [10] to explore competition between *continuous double auction markets* [11–13]. In a spirit similar to our work, they use simple learning algorithms such as Zero-Intelligence [14] or Zero-Intelligence-Plus [15] for both markets and traders, and analyse the allocation efficiency of double auction markets when they are competing against each other. Multi-agent-based simulations have mostly been used in this context and allow additional layers of complexity such as *adaptive markets* and *heterogeneous agents* to be added. We pursue instead a modelling approach that strips out as much detail as possible [6–8] to allow for detailed theoretical analysis, which can often reveal features that would be missed when relying exclusively on numerical simulations. In this spirit, while the market mechanisms implemented in the JCAT library are *continuous* double auctions, we use in our model a mechanism more similar to a *clearing house* where the clearing process takes place at discrete time steps. This makes a largely analytical approach possible, which reveals the learning process of the agents as the main driver of fragmentation. This conclusion was shown in [6] to carry over to models with more complex market mechanisms and more sophisticated agent strategies, based e.g. on [16].

Authors such as Ellison *et al.* [17] and Shi *et al.* [18] have focused on studying the competition between markets and the conditions under which this led to multiple market coexistence or the emergence of a market monopoly. The authors name two significant effects in the competition of double auctions, one of them is the positive size effect, i.e. agents prefer trading in a market where there are already many traders of the opposite type (e.g. sellers like trading at markets where there are many buyers), as the choice among offers is better. The authors additionally suggest the existence of a negative size effect in a double auction market, as agents will prefer being in the minority group to trade more often (e.g. buyers see the benefit of trading at a market where there are not many buyers, e.g. [19]). Ellison *et al.* [17] point out that due to this negative size effect, coexistence of many markets is possible. On the other hand, Shi *et al.* [18] investigate which of the two effects is stronger and finds that due to more substantial positive effects, a monopoly will in many situations be the preferred outcome. When there is strong market differentiation, Shi *et al.* [18] argue that market coexistence is

possible, especially for markets that have different pricing policies, e.g. where one market charges a fixed participation fee while another charges a profit fee. Although in what follows we will consider markets without fee charging policies, we will find nonetheless there are system parameter ranges that enable coexistence, where markets are populated by roughly the same numbers of traders; conversely, we also identify the parameter regimes for which one market is dominant. It is important to note that the studies cited above have focused on finding either the Nash equilibria or states favoured by the replicator dynamics. By contrast, we consider dynamics based on agents learning to improve their market choosing strategy, which we believe is more appropriate in the context of agents engaging in economic interactions. In this study, we show that fragmentation can arise even in an initially homogeneous population of traders, only because the traders adapt to their past record of successful trades.

# 2. Agent-based model

Here we summarize the basic assumptions and properties of the model introduced in [5,6,8] and extend it to include multiple markets.

## 2.1. Traders

We study a population of agents without sophisticated trading strategies, essentially zero-intelligence traders [14,20,21]. The orders to buy at a certain price (bids) and orders to sell at a certain price (asks) are assumed to be unrelated to previous trading success or any other information. We assume that bids, $b$, and asks, $a$, are normally distributed ($a \sim \mathcal{N}(\mu_a, \sigma_a^2)$ and $b \sim \mathcal{N}(\mu_b, \sigma_b^2)$), where $\mu_b > \mu_a$, in line with [6]. After each round of trading each agent receives a score, reflecting their payoff in the trade. The scores of agents who do trade are assigned as elsewhere in the literature [14,22]: buyers value paying less than they offered ($b$), and so their score is $S = b - \pi$, where $\pi$ is the trading price. Sellers value trading for more than their ask ($a$), and so $S = \pi - a$ is a reasonable model for their payoff.

## 2.2. Markets

The role of a market is to facilitate trades so we define markets in terms of their price-setting and order-matching mechanisms. We consider a single-unit discrete time double auction market where all orders arrive simultaneously and market clearing happens once every period after the orders are collected. We also assume that a uniform price is set by the market—once all orders have arrived, these are used to determine average bid $\langle b \rangle$ and average ask $\langle a \rangle$ and then set a global trading price in between the two

$$\pi = \langle a \rangle + \theta(\langle b \rangle - \langle a \rangle), \tag{2.1}$$

where $\theta$ fixes the price closer to the average bid ($\theta > 0.5$) or the average ask ($\theta < 0.5$); the parameter $\theta$ thus represents the bias of the market towards sellers (they earn more when $\theta > 0.5$) or buyers (earn more when $\theta < 0.5$).[1] Once the trading price has been set, all bids below this price, and all asks above it, are marked as invalid orders that cannot be executed at the current trading price. The remaining orders are executed by randomly pairing buyers and sellers; the execution price is $\pi$. Note that we assume here that each order is for a single unit of the good traded.

The most efficient resource allocation happens when demand equals supply, i.e. at the equilibrium trading price. In a set-up like ours where the bids and asks are Gaussian random variables with equal variances ($\sigma_a = \sigma_b$) and when the number of buyers is equal to the number of sellers at a given market, the equilibrium trading price corresponds to $\theta = 0.5$, i.e. the price is $\pi^{eq} = (\langle b \rangle + \langle a \rangle)/2$. We start off below by considering such efficient markets and will also call these *fair* as $\theta = 0.5$; later we allow for the possibility that markets are not fair and set the price closer to the average bid or ask ($\theta \neq 0.5$).

## 2.3. Learning rules

Agents trade repeatedly in our model, and they adapt their preferences for the various choices at their disposal from one trading period to the next. We assume that each agent decides where to trade

---

[1]Note that traders are not informed about these market biases, nor the market mechanism in general; they only obtain information through the scores they receive.

(which of many markets) at the beginning of each trading period, only based on his or her past experience. To formalize this we introduce a set of attractions $A_m$ for each player, one for each market $m = 1, 2, 3$. The attractions will generally differ from player to player, but we suppress this in the notation for now. The attractions are updated after every trading period, $n$, using the following reinforcement learning rule (similar to Q-learning [23] and the experience-weighted attraction rule [24,25])

$$A_m(n+1) = \begin{cases} (1-r)A_m(n) + rS_m(n) & \text{if the agent chose market } m \text{ in round } n \\ (1-r)A_m(n) & \text{otherwise.} \end{cases} \quad (2.2)$$

The quantity $S_m(n)$ is the score gained trading at market $m$ in the $n$th trading period. The length of the agents' memory is set by $r$: effectively an agent takes into account a sliding window of length of order $1/r$ for the weighted averaging of past returns.

Once each preference is updated, traders use the *multinomial logit function* to choose at which market to trade in the next round

$$P(M = m) = \frac{\exp(\beta A_m)}{\sum_{m'} \exp(\beta A_{m'})}. \quad (2.3)$$

This is inspired by the experience-weighted attraction literature [24,25], where $\beta$ is the *intensity of choice* and regulates how strongly the agents bias their preferences towards actions with high attractions. For $\beta \to \infty$, the agents choose the option with the highest attraction, while for $\beta \to 0$ they choose randomly with equal probabilities among all options.

Agents randomly take the role of buyer or seller in each trading round: they act as buyers with probability $p_B$, which we call their buying preference. We will study a population of traders consisting of two classes of agents with fixed buying preferences $p_B = p_B^{(1)}$ and $p_B = p_B^{(2)}$, respectively. The attractions of agents from different classes will be denoted by $A_m^{(c)}$ with $c \in \{1, 2\}$.

We will frequently study a set-up with symmetric markets (i.e. $\theta_1 = 1 - \theta_2 < 0.5$) and a population consisting of two symmetrically biased classes (i.e. $p_B^{(1)} = 1 - p_B^{(2)} > 0.5$). The setting considered as default in [6] is $(\theta_1, \theta_2, p_B^{(1)}, p_B^{(2)}) = (0.3, 0.7, 0.8, 0.2)$. It is such that the class 1 (buyers) prefer trading at market 1, that is biased to award buyers with higher returns, while agents of class 2 (sellers) prefer market 2. It has been shown previously that for low intensity of choice $\beta$, the unique fixed point of the learning dynamics is such that agents develop a higher attraction to the market that is better for them; nonetheless, they trade largely at random because of the low $\beta$. When $\beta$ is increased, this fixed point becomes unstable as buyers and sellers would congregate in different markets and so lose many trading opportunities. Instead the population fragments: agents of both classes self-organize to divide into two groups within each class. One of these groups is return oriented (e.g. buyers at market 1) and the corresponding agents earn more per single trade; the other group can be characterized as volume oriented (e.g. sellers at market 1), earning less per trade but having the opportunity to trade more often.

## 2.4. Numerical simulations

To motivate the use of this stylized model of agents choosing between multiple markets, we start with multi-agent simulations of the system. We look at a default population of traders consisting of two classes—some tend to act more as buyers ($p_B = 0.8$), others more as sellers ($p_B = 0.2$). These traders choose between three markets that differ in their biases $\theta$. We show an example of three qualitatively different distributions of the attractions of the agents in figure 1. To facilitate the interpretation of these distributions, we mark by coloured regions in each panel which market an agent prefers at the given attraction (differences), i.e. which market s/he chooses with the highest probability.

We now give a brief description of the attractions distributions in each of the panels and explain the difference between (i) strong fragmentation, which persists in the large memory limit, and (ii) weak fragmentation, which disappears in the same limit; similar results for two market systems are discussed in [6,8]. In figure 1a, one sees that the distribution of attractions has three peaks, all of which have a size of order $O(1)$ and correspond to subpopulations of traders who choose to trade mainly at a single market. In other words, the trader population (in the class shown in the figure) splits into three subpopulations that are more attracted to one market over the others, e.g. traders develop individual loyalties to one of the markets. Such distributions of attractions with more than one peak with a size of order one are called *strongly fragmented* [8]. As discussed in previous works, this does not mean the traders' preferences are frozen: they do change their preferred market but only

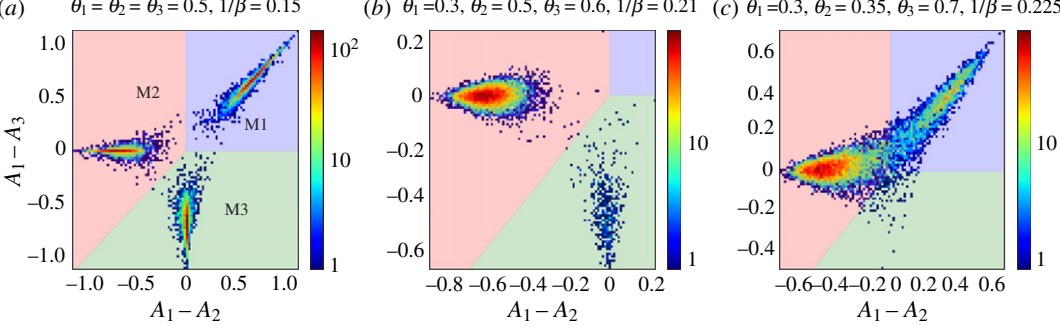

**Figure 1.** Distribution of attraction differences of population of traders for market and learning parameters as indicated in each graph title. In (*a*), the population is strongly fragmented into three groups of equal size. In (*b*), the population is weakly fragmented, the distribution has two peaks: one large peak and one peak that (as we will later see) becomes exponentially small as the memory length increases. In (*c*), the population is strongly fragmented, but only across two markets. To obtain those graphs, we ran simulations with $r = 0.01$ and $N/2 = 10\,000$ traders in each class until a steady state was reached. Traders from class 1 have preference to buy $p_B^{(1)} = 0.8$ and traders from class 2 have preference to buy $p_B^{(2)} = 0.2$. The $(A_1 - A_2, A_1 - A_3)$ plane is shown subdivided into three zones that indicate which market an agent with the corresponding attractions chooses most often. The zones are coloured blue, red and green for markets 1, 2 and 3, respectively, as indicated in (*a*).

after a long persistence time [6]. We also note that in the state shown, i.e. for the given parameters, three identical markets coexist and receive an equal share of traders, on average.

The second distribution, shown in panel (*b*), corresponds to a population divided into two loyalty groups but with different sizes: one large (order $N$) subpopulation is attracted to the second market (the fair market, $\theta = 0.5$), while the second, smaller subpopulation persistently tries to trade at market 3. The size of the smaller peak in the attraction distribution decreases exponentially as $r \to 0$ [7,8], and although markets 2 and 3 coexist for any finite $r$, in the large memory limit, market 2 has a monopoly. When attraction distributions are multimodal but only one peak has a weight of order 1 (i.e. fragmentation is only present at finite $r$) we call them *weakly fragmented*.

The distribution plotted in panel (*c*) corresponds to a strongly fragmented population, but contrary to the case depicted in panel (*a*) the third market has now lost the competition. Additionally, the share of attracted traders is not the same between the markets (as in panel (*a*)), but both peaks persist in the long memory limit.

The above simulation results offer a glimpse into a rich variety of qualitatively different structures of the attraction distributions (number and size of peaks) and consequently different outcomes of a three-market competition. To study these in more detail, we focus on the analytical and numerical methods described previously [7,8] for large populations of traders and in the large memory limit ($r \to 0$).

## 3. Analysis

To proceed with the analysis, in line with our earlier studies [6–8], we start from the fact that the system is Markovian and accordingly the master equation introduced in [6] is an exact and complete description of the evolution of agents in the limit of an infinite population $N$ and large memory $1/r$. We focus here on the steady states of this dynamical evolution. For a population with fixed buy/sell preferences, this is specified by a steady-state distribution $P(\mathbf{A}|p_B)$ where $\mathbf{A}$ is an $M$-dimensional vector of attractions and conditioning on the buying preference and distinguishes the different classes of traders. When we study more than two markets the distribution is multivariate, though we can introduce attraction differences and look for a solution in the resulting $M - 1$ variables. The master equation describing the evolution of the system [6] across the different trading rounds $n$ is not a standard linear Chapman–Kolmogorov equation as the transition kernel $K$ depends on the trading probabilities, which in turn depend on $P_n(\mathbf{A}|p_B)$. This self-consistent nature of the description arises from the reduction from a description in terms of the attractions of all $N$ agents to one for a single agent; this reduction becomes exact for $N \to \infty$. In principle, a steady state could then be found by tracking the evolution in time from the initial condition $P_0(\mathbf{A}|p_B) = \delta(\mathbf{A})$, which corresponds to all agents having zero attraction to all markets. We take a different route and first transform the time evolution equation to a Fokker–Planck description using the Kramers–Moyal expansion. This is appropriate for small $r$, i.e. for agents with long memory.

Even after the simplification to a Fokker–Planck equation, the dimensionality of the problem makes finding the steady state a non-trivial task. But we can make progress by considering the limit $r \to 0$; this will allow us to evaluate the onset of fragmentation. We do this by analysing the drift $\mu_m^{(c)}$ in the Fokker–Planck equation, defined in appendix A. To find the single agent steady state, we will search for zeros of the drift assuming fixed market order parameters, i.e. trading probabilities. We start by assuming that the two classes have homogeneous preferences for the markets (i.e. $P(\mathbf{A}^{(c)} | p_{\mathcal{B}}^{(c)})$ is a delta distribution). This is the expected solution in the low $\beta$-limit, when the steady state is unfragmented. With this assumption, the expressions for the market order parameters simplify, and we can solve the simultaneous equations for the two classes. At any fixed point solution $(\mathbf{A}^{(1)^*}, \mathbf{A}^{(2)^*})$ we evaluate the market order parameters and check if the single agent dynamics is consistent with the homogeneous population assumption: when we solve $\mu_m^{(c)}(\mathbf{A}) = 0$ we expect only one zero that coincides with $(\mathbf{A}^*)$. The onset of fragmentation (weak or strong) is then given by the intensity of choice where the single agent dynamics first has multiple zeros when evaluated at the homogeneous population market order parameters, which indicates that for $r > 0$ the distribution of attractions will have multiple peaks. To find the weights of the attraction distribution at each peak, corresponding to a fixed point, we use the Freidlin–Wentzell approach detailed in appendix B. This allows us to differentiate between small peaks, which decay exponentially with the memory length $1/r$, and large peaks, whose weight remains finite and of order unity when the $r \to 0$ limit is taken.

In the rest of the paper, we focus our analysis on a scenario with $M = 3$ markets and we describe each of the two classes in terms of the two attraction differences $\Delta A_2 = A_1 - A_2$ and $\Delta A_3 = A_1 - A_3$. We perform a Kramers–Moyal expansion of the trader's learning dynamics and obtain two Fokker–Planck equations (one for each class $c \in \{1, 2\}$ of traders) for the distribution of attraction differences $P(\Delta \mathbf{A}^{(c)}, t)$

$$\partial_t P(\Delta \mathbf{A}^{(c)}, t) = -\sum_{m=2}^{3} \partial_{\Delta A_m^{(c)}} [\mu_m^{(c)}(\Delta \mathbf{A}^{(c)}, f_1, f_2, f_3) P(\Delta \mathbf{A}^{(c)}, t)]$$

$$+ \frac{r}{2} \sum_{m,m'=2}^{3} \partial_{\Delta A_m^{(c)}} \partial_{\Delta A_{m'}^{(c)}} [\Sigma_{mm'}^{(c)}(\Delta \mathbf{A}^{(c)}, f_1, f_2, f_3) P(\Delta \mathbf{A}^{(c)}, t)]. \qquad (3.1)$$

Here the time variable $t = nr$ is a rescaled number of trading rounds, $\Delta \mathbf{A}^{(c)} = (\Delta A_2^{(c)}, \Delta A_3^{(c)})$ and $f_m$ is the market order parameter, i.e. the ratio of buyers to sellers at market $m$ (effectively the demand-to-supply ratio). The expressions for the drift vectors $\mu_m^{(c)}(\Delta \mathbf{A}^{(c)}, f_1, f_2, f_3)$ and the covariance matrices $\Sigma_{mm'}^{(c)}(\Delta \mathbf{A}^{(c)}, f_1, f_2, f_3)$ for each class are given in appendix A.

## 3.1. Three fair markets

We start by looking at what happens when the three markets available are all fair, i.e. $\theta_1 = \theta_2 = \theta_3 = 0.5$. This means they set their trading price to be exactly the mean of the average bid and the average ask. As mentioned previously, the fair market corresponds to a market mechanism delivering the equilibrium trading price, provided the number of buyers equals number of sellers.

Based on intuition from similar physical systems, one might expect spontaneous symmetry breaking, where random fluctuations lead the whole population to select only one of the possible symmetric markets. However, in stochastic multi-agent simulations we observe instead steady states with fragmented populations within each class; we therefore focus on steady states of the traders' learning dynamics without symmetry breaking.

Since the three markets have the same bias $\theta$, in a symmetric solution, they should attract the same number of agents, irrespective of their class. On the other hand, as we study classes of agents with symmetric preferences to buy $p_{\mathcal{B}}^{(1)} = 1 - p_{\mathcal{B}}^{(2)}$, the difference between the number of buyers and the number of sellers at a single market is of order $\sqrt{N}$, $N_{\mathcal{B}} = N_{\mathcal{S}} + O(\sqrt{N})$. As a consequence, in the large size limit, the ratio of the number of buyers to the number of sellers in each market is equal to 1. This simplification is the reason why we choose to start the analysis with the simple case of three fair markets, which allows one to explore the phenomenon of fragmentation across three double auction markets without the need for a self-consistent determination of market order parameters [7,8].

We start by looking at the fixed point structure of the single agent dynamics when the intensity of choice $\beta$ is small. As expected, the only fixed point of the learning dynamics is $A_1 - A_2 = A_1 - A_3 = 0$ and corresponds to a trader who chooses to randomize between the three markets (figure 2a). When the intensity of choice $\beta$ reaches a critical value $\beta_c = 1/0.254$, three saddle node bifurcations take place simultaneously and three pairs of stable and unstable fixed points appear (figure 2b). The reason why

deep

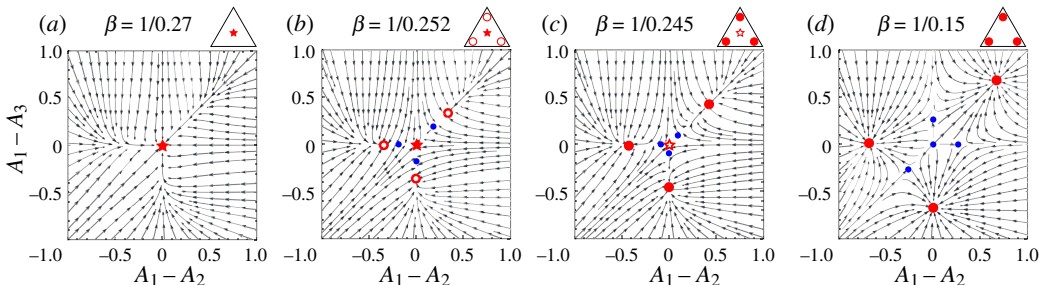

**Figure 2.** Flow diagram and fixed points of the learning dynamics of a single trader with $p_B^{(2)} = 0.2$, choosing between three fair markets. (*a*) Below the weak fragmentation threshold $\beta = 1/0.254$, the dynamics only has one fixed point, which is stable (denoted with red star). (*b*) When $\beta$ reaches the weak fragmentation threshold $\beta_c = 1/0.254$, three pairs of unstable (blue) and meta-stable (red empty circles) fixed points appear and the system becomes weakly fragmented with one large peak, which corresponds to traders randomizing between the three markets, and three small peaks where agents trade preferentially at one of the three available markets. (*c*) At $\beta_c' = 1/0.252$, the three outer fixed points become stable and the central one meta-stable and the system is now strongly fragmented, with three peaks of equal size each of which corresponds to preferentially trading at a single market. (*d*) As $\beta$ increases, the meta-stable fixed points eventually becomes unstable. Above each graph we indicate in triangular notation the category to which each of the fixed point structures belongs (see main text for details).

those three saddle node bifurcations take place at the same time lies in the markets' symmetry, i.e. their identical bias $\theta = 0.5$. In the more general case where the three markets are different, we expect the appearance of each pair of new fixed points to take place at a different value of $\beta$.

When looking at the deterministic dynamics for low intensity of choice (figure 2*a*), it is obvious that the system is not fragmented and there is only one stable fixed point. At larger intensities of choice as in figure 2*b*–*d*, knowing the deterministic dynamics is not sufficient to distinguish between 'stable' fixed points (the ones where, in our terminology, large peaks will be centred) and 'metastable' ones (which for us indicate the positions of the small peaks). To assess the stability of fixed points in figure 2 and weight sizes of potential peaks, we use the Freidlin–Wentzell approach detailed in appendix B.

As an example of an attraction distribution that has both small and large peaks we consider the range $1/0.252 \geq \beta \geq 1/0.254$ for the intensity of choice, where the system is weakly fragmented (as in figure 2*b*). The central fixed point is stable and a large peak in the attraction distribution is located at this fixed point, while the three outer fixed points are metastable and correspond to small peaks. As $\beta$ is increased to a second critical value of $\beta_c' = 1/0.252$, the three outer fixed points become stable and the system undergoes a strong fragmentation transition. For any values of $\beta$ above this second fragmentation threshold, the system will be strongly fragmented as the distribution of preferences of the traders will have three peaks of equal weight, each of which corresponds to a stable fixed point of the single agent dynamics (red points in figure 2*c*,*d*). For $1/0.237 \leq \beta \leq 1/0.252$, the distribution of attractions retains an additional peak at the fixed point at $(0, 0)$ but the weight of this peak will become exponentially small as the memory length increases (figure 2*c*). This metastable fixed point and the associated small peak in the attraction distribution then disappear for $\beta \geq \beta_c'' = 1/0.237$ (figure 2*d*).

We summarize briefly the intuitive meaning of the above results for the attraction distributions in a system of agents with long memory choosing between three fair markets. When the intensity of choice is small the agents cannot develop strong attractions to any particular market as low $\beta$ implies that they choose a market largely randomly. With increasing $\beta$, three small subpopulations of the agents in each class develop a loyalty to one of the markets, signalled by increased attractions, but the random choice strategy remains dominant. These loyal subpopulations grow until (beyond $\beta_c'$) they encompass most of each agent class.

To help with understanding the variety of different steady states, we introduced an attraction distribution notation in the shape of triangles, as depicted in panels of figure 2. We focus on the number and size of the peaks, rather than their exact position, and use the triangle to visualize attraction to any of the three markets (circle close to the corner) or market indifference (star shape). To distinguish between large and small peaks we use filled or empty objects (both stars and circles).

In the simple case of three competing markets considered so far, we find that they always coexist, but in different scenarios ranging from all traders choosing a market randomly to traders splitting into subpopulations with persistent market loyalties. An obvious question is then whether this

fragmentation is critically dependent on the fact that all the markets are identical. To answer this, we next extend our analysis to markets with different biases.

# 4. Exploration of the parameter space: markets with different biases

Each market bias $\theta_1$, $\theta_2$, $\theta_3$ is between zero and one, i.e. the market parameter space is a unit cube. Of course the phenomenon of fragmentation is independent under permutation of the market biases as this effectively just changes the labelling of the markets. We can therefore restrict our analysis to 1/6 of the cube where $\theta_1 \leq \theta_2 \leq \theta_3$ and can reconstruct the behaviour in the rest of the parameter space by symmetry. We will mostly follow this scheme but sometimes allow a different parameter ordering to get simpler two-dimensional phase diagrams, with a typical bias along the x-axis and the inverse intensity of choice along the y-axis. We study three different types of scenarios, guided by explorations in our previous work: (i) one fair market $\theta_2 = 0.5$ and two symmetrically biased markets $\theta_1 = 1 - \theta_3$, with $\theta_1$ as a free parameter varying between 0 and 1/2, shown in figure 3, (ii) two symmetrically biased markets $\theta_1 = 0.3$, $\theta_2 = 0.7$ with $\theta_3$ varied as a free parameter, shown in figure 4, (iii) $\theta_1 = 0.3$, $\theta_2 = 0.5$ and $\theta_3$ again ranging from 0 to 1, shown in figure 6. As will become clear in the rest of this section, these parameter settings allow for the analysis of the effect of a number of properties on the occurrence of fragmentation, such as the market symmetry, the 'distance' between market biases and the effect of market fairness.

## 4.1. Two symmetrically biased markets and one fair market

Following the reasoning we used in the case of three fair markets, we continue to focus on solutions that do not break the market symmetries. This assumption is supported by stochastic multi-agent simulations in which we do not observe market symmetry breaking. We use the symmetries to restrict the possible values of the 'market aggregates', i.e. the demand-to-supply ratios. In particular, we can show that these ratios are inverses of each other for the symmetrically biased markets, and that the ratio is unity at the fair market as before. To see this, note first that when $\theta_1 = 1 - \theta_3$ and $\theta_2 = 0.5$, for traders with symmetric preferences to buy, the role played by market 1 for traders from class 1 is the same as the role played by market 3 for traders from class 2 and vice versa. As a consequence, the probability of trading at the first market for a trader from class 1 (resp. 2) is equal to the probability of trading at the third market for a trader of class 2 (resp. 1). We can write the buyer/seller ratios in market 1 and 3 as

$$
f_1 = \frac{P^{(1)}(M=1)p_B^{(1)} + P^{(2)}(M=1)p_B^{(2)}}{P^{(1)}(M=1)(1-p_B^{(1)}) + P^{(2)}(M=1)(1-p_B^{(2)})}
$$

and

$$
f_3 = \frac{P^{(1)}(M=3)p_B^{(1)} + P^{(2)}(M=3)p_B^{(2)}}{P^{(1)}(M=3)(1-p_B^{(1)}) + P^{(2)}(M=3)(1-p_B^{(2)})}.
$$

(4.1)

When substituting into these expressions the equalities $P^{(1)}(M=1) = P^{(2)}(M=3)$, $P^{(2)}(M=1) = P^{(1)}(M=3)$ and remembering that $p_B^{(1)} = 1 - p_B^{(2)}$, one sees that $f_1 = 1/f_3$. The fact that the ratio of buyers to sellers at the fair market (market 2) is unity follows by analogous reasoning.

Let us first calculate the value of the intensity of choice at which traders start to fragment weakly. To do so, for a given value of the free parameter $\theta_1$, we start from low values of $\beta$ and gradually increase the intensity of choice until it reaches a critical value where the single agent dynamics has two stable fixed points. Those values of $\beta$ are shown by the upper solid line in figure 3.

The natural continuation of this analysis is to look—if it exists—for the strong fragmentation threshold. While thanks to our previous analysis of symmetric markets we know that for $\theta_1 = 0.5$ strong fragmentation takes place at $\beta = 1/0.252$, our numerical methods show that for reasonably asymmetric markets, i.e. $\theta_1 < 0.48$, strong fragmentation does not take place across the entire range of values of $\beta$ that we consider numerically for our phase diagram. For $\theta_1$ between 0.48 and 0.5, our numerics suggest possible strong fragmentation but a definite conclusion cannot be reached given the numerical precision limits of the required action minimizations.

To distinguish between different types of steady states in the following analysis—the number of emergent loyalty groups, their market preferences and sizes, we now introduce a triangle notation that is illustrated in figure 2 and used in the ($\theta_1$, $1/\beta$) phase diagram there. Each of the triangle corners represent preferences for one of the three markets, while full and empty circles represent large/small peaks; different colours denote the different trader classes. This notation allows us to

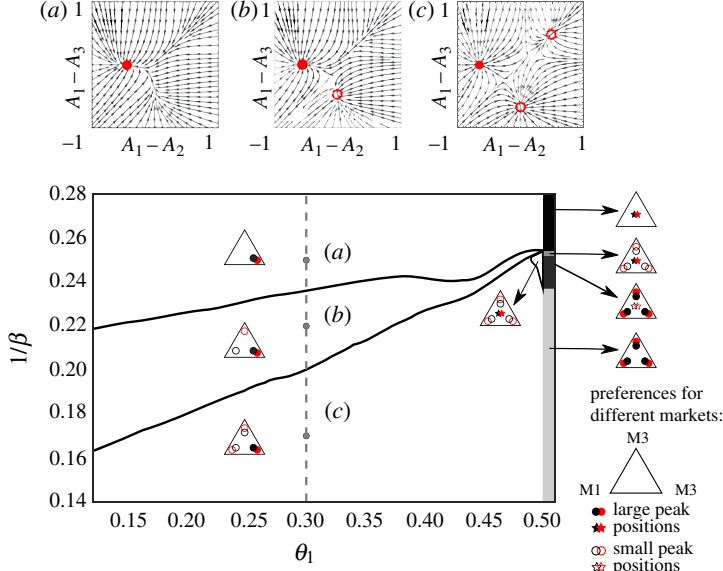

**Figure 3.** Peak structure of the steady-state distribution of traders' preferences when they learn to choose between three markets, two of which have symmetric market biases $\theta_1 = 1 - \theta_3$ and one of which is fair. The three insets on the top show the fixed point structure for an agent from class 2 ($p_{\mathcal{B}}^{(2)} = 0.2$), for $\theta_1 = 0.3$ and different $\beta$ as indicated in the phase diagram by grey points. Full circles in the phase diagram correspond to a stable (large peak) and empty circles to a metastable fixed point (small peak), colours (black = class 1, red = class 2) differentiate between the agent classes. The grey band at $\theta_1 = 0.5$ shows the type of attraction distribution for the case $\theta_1 = 0.5$, i.e. when the three markets are fair; examples of attraction distribution peak structures in that region are shown in figure 2.

quickly realize whether some markets lost the competition, which markets are dominant, and which might attract only a single class of traders. Additionally, we use a star to denote an attraction distribution peak without preferences for a specific market. This is present only for the scenario with three fair markets, as depicted in the right band of the phase diagram in figure 4. The triangular representations shown on the right correspond to the flow diagrams with fixed points depicted in figure 2.

In figure 3, we see that for any value of $\beta$ and $\theta_1 < 0.5$, the majority of the traders will prefer to trade at the fair market (market number two), so that this market will have a monopoly in the $r \to 0$ limit. When agents have finite memory, all three markets coexist when $\beta$ is greater than the weak fragmentation threshold, but market 2 still attracts the majority of trades. Interestingly, in the region of the phase diagram with intermediate $\beta$ (see inset (b)), all three markets coexist, but markets 1 and 3 are visited by only a single class, despite the fact that trading opportunities are lower that way.

In summary, the results depicted in figure 3 tell us that, apart from the particular case when the three markets are all fair, *strong* fragmentation does not take place when a fair market competes against two symmetrically biased markets. We therefore move next to an even less symmetric situation.

## 4.2. Two symmetric markets and one biased market

We continue exploration of the space of market biases by considering two symmetric markets with fixed market biases $\theta_1 = 0.3$ and $\theta_3 = 0.7$; this is the market set-up we mostly studied in previous works. Without the third market, when the two classes of traders adaptively choose between two symmetric markets one finds both weak and strong fragmentation above $\beta_c = 1/0.28$ [8]. Here, we add the third market and vary its bias, which as figure 4 shows leads to a range of different steady-state attraction distributions.

We first note that strong fragmentation appears, and does so across a reasonably broad range of market biases (grey zone in figure 4). This range excludes the case studied above where market 2 is fair: strong fragmentation occurs only for $\theta_2 \notin [0.45, 0.55]$, i.e. when the second market is sufficiently biased. For $\theta_2 < 0.45$ (resp. $\theta_2 > 0.55$) the traders from the first (resp. second) class *strongly fragment* across the two markets that maximize average profit per trade for each class. For example, in the case of $\theta_2 = 0.4$, buyers (traders in class 1, who have $p_{\mathcal{B}}^{(1)} = 0.8$) will prefer trading at markets 1 and 2 while the sellers remain unfragmented.

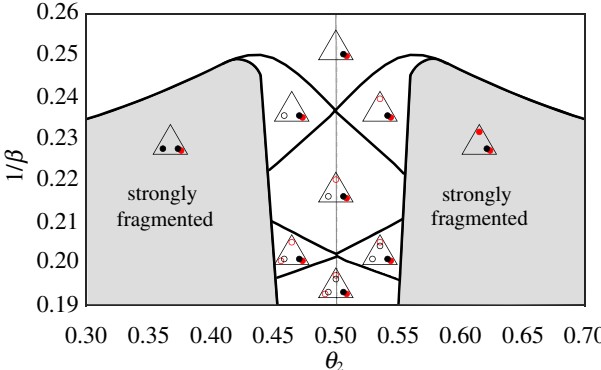

**Figure 4.** Types of attraction distributions in the population choosing between markets $\theta_1 = 1 - \theta_3 = 0.3$ and varying $\theta_2$. The grey zone indicates the region in parameter space where the distribution of attractions has two large peaks for at least one class of agents, i.e. where strong fragmentation occurs. Note that between every unfragmented and strongly fragmented region (appearance of large loyalty groups at market 1 and 3) there is always a weakly fragmented region (where the same loyalty group, i.e. peak in the distribution, is small), but these regions are mostly too narrow to be visible. The grey line in the centre corresponds to the dashed line in figure 3.

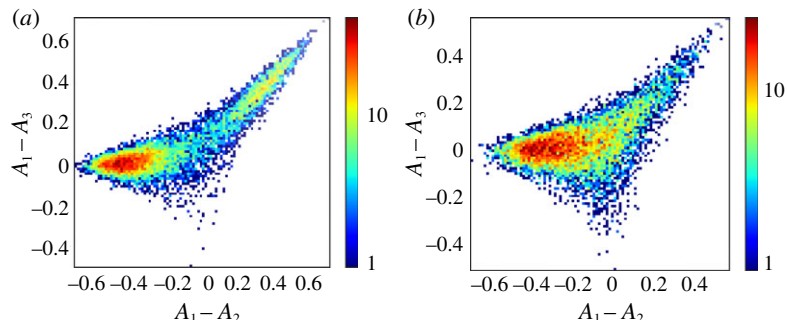

**Figure 5.** Distribution of attraction differences of traders who choose between three markets with market biases $(\theta_1, \theta_2, \theta_3) = (0.3, 0.35, 0.7)$. The population consists of two classes of $N/2 = 10^4$ traders with symmetric buy-sell preferences $p_\mathcal{B}^{(1)} = 1 - p_\mathcal{B}^{(2)} = 0.8$, inverse memory length $r = 0.01$ and intensity of choice $\beta = 1/0.21$. We see that the attraction distribution of the first class is strongly fragmented (panel ($a$)), while the second one is unfragmented (panel ($b$)), as predicted by the phase diagram in figure 4.

We do not explore the phase diagram below the first strong fragmentation threshold as this would require the numerical solution of self-consistency conditions for multiple aggregates in the presence of two (or more) strong fragmentation peaks in the traders' attraction distributions. This is numerically very challenging and so we leave it for future work. However, it is possible to get an intuition about the shape of the phase diagram below this threshold by extrapolating the zones of weak fragmentation in the range of $\theta_2$ where the second market is close to fair.

We show in figure 4 graphically the types of steady state attraction distribution within the different regions of the phase diagram. These predictions are obtained using single agent flow diagrams as shown in figures 2 and 3. We show an exemplary comparison to stochastic multi-agent simulations in figure 5 and find excellent qualitative agreement. The agent class that mostly buys (class 1, left panel) fragments into two subpopulations mainly trading at markets 1 and 2, respectively, where they maximize their profit because $\theta_1, \theta_2 < 0.5$. Agents in the class that mostly sells prefer market 2 as the less biased of the two markets that are populated by the buyers. We conjecture that it is the asymmetry imposed by two markets favouring buyers that leads to a consolidation around markets favouring buyers, while sellers do not develop attractions toward the market that favours them.

Having described the range of values of $\theta_2$ for which strong fragmentation takes place, we inspect more closely the range of parameters for which only weak fragmentation occurs (figure 4). To do so, we look at how the attraction distributions of both classes of traders evolve at fixed $\theta_2 = 0.47$ when $\beta$ increases. For values of $\beta$ small enough in relation to the agents' attractions, they will essentially randomize their market choice, with a weak preference towards the market that is closest to fair, market 2. This preference increases with $\beta$ so that traders from the two classes effectively coordinate

at market 2, providing a good trade-off between profit and trading volume. As $\beta$ grows further, additional small peaks arise in the attraction distributions while most of the traders remain in the fairer market. In particular, at $\beta = 1/0.246$ a peak corresponding to the strategy 'trading at the profit maximizing market' (market 1, which has $\theta_1 = 0.3$) appears for class 1. Then at $\beta = 1/0.228$, a peak corresponding to the strategy 'trading at the profit maximizing market' (market 3 with $\theta_3 = 0.7$) appears in the attraction distribution of the agents from the second class. After those two successive appearances of weak fragmentation between the fairer market and the profit maximizing market for both class 1 and class 2, further peaks in the attraction distribution—which correspond to the strategy 'trading at the volume maximizing market'—appear successively for class 2 at $\beta = 1/0.207$ and then for class 1 at $\beta = 1/0.198$.

Our phase diagram suggests that fairness of the second market weakens fragmentation. We cannot exclude, however, that strong fragmentation might occur even for $\theta_2$ close to 0.5, for larger $\beta$ (lower $1/\beta$) than investigated in the phase diagram of figure 4.

Interestingly, addition of the third market leads to trade shifting away from one of the symmetric markets, throughout the entire strong fragmentation region in figure 4. Only when the added market is close to fair can the two symmetric markets continue to coexist, though with both receiving only a small fraction of trades. Market 2 in fact has the largest market share throughout figure 4.

We can summarize the intuition behind the above results as follows. As the intensity of choice increases, each class of agents will first fragment weakly between a market that is close to fair (market 2) and the market that maximizes profit for them, and then fragment weakly across all three markets. On the other hand, if the second market is not fair, the class for which this market is more profitable will fragment strongly between their two profit maximizing markets, while the other class will only trade at the market that is closest to fair. The results of this subsection suggest that as soon as traders have at their disposal a reasonably fair market, they are not going to fragment and will prefer to trade with the fair market; when they have no fair market they will always prefer the profit maximizing market, and will visit the volume maximizing market (which brings lower profits but typically more trades) only as a last resort.

## 4.3. Markets without symmetry

The two examples presented in §§4.1 and 4.2 lead to the conjecture that the presence of a fair or nearly fair market—which provides a good trade-off between profit in individual trades and trading volume—can suppress fragmentation. To confirm this conjecture, we consider three markets where the first one is biased toward buyers ($\theta_1 = 0.3$) and the second one is fair ($\theta_2 = 0.5$); the bias of the third market is the parameter we will vary.

As we did in the previous subsections, we will draw a phase diagram of the type of attraction distribution for the two agent classes, as a function of the intensity of choice $\beta$ and the bias of the third market $\theta_3 \in [0, 1]$. The result in figure 6 shows that within the range of parameters explored, if there is fragmentation it is weak, so that the attraction distributions for both trader classes always become unimodal in the $r \to 0$ limit. (Extrapolation to lower $1/\beta$ than shown in figure 6 suggests that this situation does not change at even larger intensity of choice.) Only one peak has weight of order one and, depending on the values of $\beta$ and $\theta_3$, the steady state is either unfragmented or weakly fragmented, having one or two small peaks that disappear in the $r \to 0$ limit.

One notes that once the intensity of choice increases above a certain threshold value shown by the full black line in figure 6, a weak peak corresponding to the strategy 'trading at market 1' appears in the distribution of attractions of the first class of agents, whose attractions are marked by black circles; recall here that market 1 provides buyers, who are more frequent among agents of the first class, with higher returns. When $\beta$ crosses the second fragmentation threshold (red line in figure 6), the same type of weak peak emerges in the distribution of attractions of the second class of agents (denoted by a red empty circle as before).

The fact that the two solid lines just described are close to horizontal reflects the fact that since almost all of the population trades at the fair market, the bias of the third market will not significantly influence the preference of traders. This is the reason why the intensity of choice at which traders of class 1 (resp. class 2) will weakly fragment between markets 1 and 2 is almost independent of the bias of the third market. The same is not true of the thresholds for the appearance of a peak corresponding to the strategy 'trade at market 3', which are indicated by the sloping dashed lines in figure 6.

Consistent with previously discussed results, the existence of fair market suppresses strong fragmentation and within the space of parameters depicted in figure 6 we note only weak

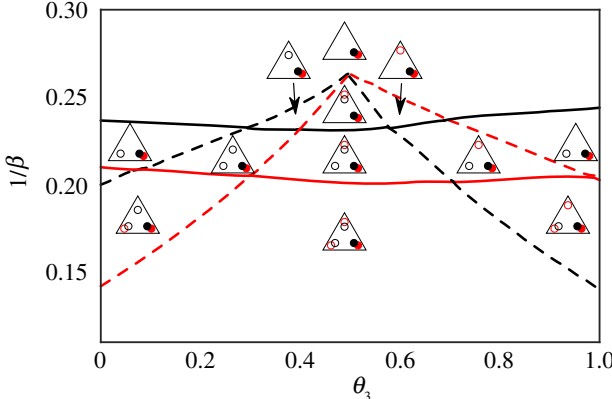

**Figure 6.** Peak structure of the different attraction distributions when $\theta_1 = 0.3$, $\theta_2 = 0.5$, $p_B^{(1)} = 1 - p_B^{(2)} = 0.8$. The solid/dashed lines show weak fragmentation transitions where subpopulations emerge that favour markets 1 or 3 (line colours denote class of agents in which transition occurs).

fragmentation. This means that across the parameter range investigated the fair market attracts most of the traders. We also note that the third market loses the competition when it is very biased and the intensity of choice is not large enough (note regions where market 3 either attracts none or only one class). However, it is interesting to see that for sufficiently large intensity of choice $\beta$ all three markets coexist independently of the third market bias.

# 5. General number of markets M

So far we have discussed various cases of fragmentation in the three-market set-up. We found that above some critical value of the intensity of choice $\beta$, the solution in which the population remains indecisive towards the markets is never stable and at least one market loyalty group is formed. The obvious question is now whether we can say something about the number of distinct agent subpopulations in the general case of $M$ markets.

The theoretical description of the population's adaptation in the most general case, without market or agent symmetry requires the self-consistent procedure of calculating order parameters (one per market) and the steady-state distribution of the agent attractions. This is a non-trivial task in higher dimensions but the general existence of solutions can be rationalized within a simple counting argument.

In the following, we make the assumption that for all $M$ there is a fragmentation threshold $\beta_s$ above which the drift in the Fokker–Planck representation of the dynamics has multiple zeros. However, even when this is the case it is not clear whether all agent classes will develop loyalty groups towards each of the markets (and the corresponding attraction distribution peaks), whether the peaks will be small or large; in the latter case fragmentation persists by definition in the $r \to 0$ limit. To address this question we consider an agent class that is strongly fragmented across $M$ markets so that in the limit $r \to 0$ its attraction distribution consists of $M$ delta peaks with weights of order unity. We can find the peak positions by locating the zeros of the drift, but without the Fokker–Planck solution, we cannot obtain the peak weights and the Freidlin–Wentzell approach becomes difficult. We therefore ask how many non-zero peak weights can exist in general, for $C$ agent classes and $M$ markets. As explained, we assume the general shape of the steady-state distribution

$$P^{(c)}(\mathbf{A}) = \sum_{m=1}^{M} \omega_m^{(c)} \delta(\mathbf{A} - \mathbf{A}_m^{(c)}).$$

Each of the agent classes is described by peak weights $\omega_1^{(c)}, \ldots, \omega_M^{(c)}$ that satisfy the normalization condition $\sum_{m=1}^{M} \omega_m^{(c)} = 1$, thus in the absence of any symmetry we have $M - 1$ free variables per class. On the other hand, for each market we define an order parameter $f_m$, thus the system of equations we need to solve to find a strongly fragmented solution is

$$\mathcal{F}_m(\omega_1^{(1)}, \omega_2^{(1)}, \ldots, \omega_M^{(1)}, \ldots, \omega_M^{(C)}) = f_m.$$

Here $\mathcal{F}_m$ denotes the relationship between the peak weights and market order parameters; an example of this for $C = 2$ and $M = 3$ is written explicitly in equation (4.1). Without symmetries, when all the equations and variables are independent, this system of $M$ equations and $C(M-1)$ variables has a unique solution only when the number of equations is equal to the number of variables, i.e. $M = C(M-1)$. This equation has an integer solution pair only when both number of market $M$ and classes $C$ is equal to two, $(C, M) = (2, 2)$. For example, the population studied so far with its $C = 2$ agent classes requires $2(M-1)$ weights for strong fragmentation across $M$ markets, and equating the number of variables $2(M-1)$ and the number of equations $M$ gives $M = 2$ markets, which is the case studied in [6].

Since we have seen that full fragmentation, with all agent classes developing separate loyalty groups for all markets, can only happen (without symmetries) in systems with two markets and two agent classes, we next relax the assumption on the number of loyalty groups. Let us suppose there are $M$ markets and two agent classes, each of them fragmenting into $\eta^{(c)}$ subgroups (i.e. having only $\eta^{(c)}$ non-zero peak weights), the system of equations for these weights has a unique solution when $\eta^{(1)} + \eta^{(2)} - 2 = M$. This shows that if one class divides into $M$ loyalty groups, the second class will fragment only across two markets; other combinations satisfying $\eta^{(1)} + \eta^{(2)} = M + 2$ are also possible. For a general number of agent classes, the analogous constraint reads

$$\eta^{(1)} + \eta^{(2)} + \cdots + \eta^{(C)} = M + C. \tag{5.1}$$

As an example, if one class develops loyalty groups to all $M$ markets, the other $C-1$ classes can have $C$ such subpopulations in total, equating to one bimodal and $C-2$ unimodal steady-state distributions. More generally, if we associate each loyalty group with its preferred market then (5.1) shows that it is impossible for the population classes to develop disjoint sets of preferred markets, as that would require $\eta^{(1)} + \eta^{(2)} + \cdots + \eta^{(C)} \leq M$. For example, in the case $C = 2$, there will be at least two markets for which both classes have loyalty groups; the overlap will be even greater if some markets lose out and have no associated loyalty group.

Summarizing, the conclusion of our counting argument is that in the $r \to 0$ limit at most $C + M$ loyalty groups can coexist. In the three-market scenario with two classes, this is at most five loyalty groups. We saw an exception in the case of three fair markets, where six loyalty groups can exist; this is because of the symmetry between the markets, which our general argument excludes. It is remarkable how the simple counting argument gives a variety of new conjectures for the systems with multiple markets. It provides a maximal number of loyalty groups; it tells us that all markets can in principle coexist, and that the loyalty groups of different agent classes must overlap at $C$ markets at least. An interesting consequence is the emergence of a state where some markets are persistently visited only by a subset of the overall population of traders.

# 6. Summary and outlook

In this paper, we have investigated whether market coexistence is possible in systems with more than two markets when agents with fixed buy/sell preferences adapt dynamically to optimize their choice of market. This research question is motivated by empirical observations of multiple markets coexisting and attracting loyal traders both in *in silico* and real market competitions. Rather than aiming to reproduce market stylized facts, here we investigate mechanisms that might lead to a previously neglected phenomenon, namely, that multiple market loyalties, and thus market coexistence, could emerge without any underlying heterogeneity of agents or markets and only as a consequence of the co-adaptation of the agents. To this end, we studied the possible steady states of the agent dynamics, in particular with regard to the occurrence of fragmentation, where a homogeneous class of agents spontaneously forms subpopulations with long-lived market preferences.

The proposed model contains an implicit assumption of bounded rationality as the agents do not optimize any utility function or aim to make the rational/optimal choice; instead their behaviour is based on their past observed outcomes. Depending on the learning parameters the agents are tunable between trading randomly and a behaviour that repeats the most rewarding past choices. The agents do not possess knowledge about market mechanisms nor the existence of various different agents nor their scores, they only make decisions based on their past observations. In this regard, these assumptions violate rational agent assumptions due to the lack of information and lack of utility-optimizing behaviour. Nonetheless, in the case of two markets it has been shown [7] that when the agents' memory is infinitely long ($r \to 0$) and they do not update their preferences for options they did

not try in the last steps, then the expected outcome under rational behaviour (Nash equilibrium) is retrieved.

Motivated by the wide variety of structures of the attraction distributions that one observes in multi-agent simulations, we explored different combinations of market biases and their influence on the phenomenon of fragmentation. First we studied fragmentation across three fair markets, i.e. with $\theta_1 = \theta_2 = \theta_3 = 0.5$. This was the only scenario where we found that all three markets coexist across the full range of the intensity of choice $\beta$ of the agents. As $\beta$ increases we nonetheless see a change, from an indecisive population (where agents visit all three markets randomly) to a strongly fragmented population where each agent class splits into three equal-sized loyalty groups with a distinct preference for one market.

We continued by exploring different market configurations to get an intuition for the factors that drive fragmentation. This enabled us to identify two principal causes of fragmentation: (i) the *similarity between the markets' biases*, (ii) the *average volume of trade and average profit earned at a market*. The *similarity* between two markets is going to enhance fragmentation because traders are more likely to split across two markets if they effectively cannot tell them apart. This effect is visible in §4.3 where the strong and weak fragmentation thresholds are the highest (in terms of $1/\beta$) when the second market and the fair market have the same bias. The ordering of the appearance of the peaks in the traders' attraction distributions suggests—as we pointed out in §4.2—that traders will have an initial preference for markets that provide a good balance between trading volume and profit, then as the intensity of choice increases they will first spread to the market that maximizes their profit and then subsequently to the one that maximizes their trading volume.

The concepts of positive and negative size effects introduced previously [17,18] are useful when thinking about traders who develop loyalty for markets that do not reward them highly. At these markets, traders benefit from the many trading options available (positive size effects), and the fact that they are in the minority group (negative size effects). However, contrary to the findings of Ellison *et al.* [17] and Shi *et al.* [18], we note that market coexistence is more prevalent when the markets are similar—the fragmentation region shrinks with increased market difference.

Apart from the case of three identical markets, we find that once $\beta$ is large enough for agents to stop choosing markets at random, the three markets never coexist fully in the large memory limit, i.e. at least one of them will have a market share that vanishes for $r \to 0$. At most, we observe that the population fragments strongly across *two* markets (see strong fragmentation in figure 4). These markets then each have a finite share of the trading volume for $r \to 0$, though with one being subdominant because it is visited only by (some of the) agents from a single class.

From a general counting argument, we found further that full market coexistence, where all agent classes develop the (joint) maximal number of loyalty groups, leads to apparently specialized markets: some agent classes develop loyalties only to a subset of all markets (as in figure 4) and conversely some markets are not visited by agents from all classes. This is not a consequence of a market explicitly targeting some subset of the agent population, but rather of the limited number of market loyalties the different agent classes can support.

We mostly considered moderate values of $\beta$ driven by our interest in finding domains of different steady states, and for those purposes our straightforward implementation of the action minimization algorithm served us well. However, for large values of $\beta$ it occasionally fails to find minimal action path, thus robustness and accuracy improvements are needed if one is interested particularly in this regime. One possibility might be to use the geometric minimum action method [26].

Although the analytical and numerical methodology we have proposed to study agents who choose between multiple markets is valid for any number of markets $M$, it is challenging for two reasons: (i) the parameter space dimension grows with $M$ thus making numerical exploration of all possible behaviours difficult, and (ii) analytical approaches also become harder to implement as the analysis is done in the space of attraction differences of dimension $M - 1$.

Turning to implications for market competition, our results show that loyalty groups for all three markets rarely exist for large intensity of choice $\beta$ in the large memory limit ($r \to 0$). However, for finite memory ($r > 0$), one should expect that the small peaks persist. In two market systems, above certain values of $r$ (effectively for short memory) only a strongly fragmented steady state exists [6] instead of two weakly fragmented and metastable strongly fragmented states; it would be interesting to investigate if similar results also hold for multiple markets.

In this and previous studies, we have investigated how agents adapt based on their exploration of markets; the adaptation mechanism implicitly assumes that markets do not change. Realistically, one would expect that a market tries to adapt as well once the number of traders using it decreases. If

markets only try to maximize this number of traders, one could speculate that by adapting their $\theta$ biases they would converge to all-fair markets (similarly to the Hotelling paradox [27]). If on the other hand markets were to adapt to optimize the number of *successful* trades, by e.g. charging fixed or profit-dependent fees, then it would be intriguing to know what types of steady states would be realized in the overall system of agents and markets.

Finally, a broad implication of our study is that fragmentation (weak or strong) can emerge spontaneously within a class of homogeneous traders, in contrast to statements elsewhere [1] arguing that heterogeneity among traders is the reason for market fragmentation. This we think is a very interesting result as it demonstrates that structure in the preferences of economic agents might emerge out of adaptation rather than being present from the start. To this end, we made an assumption of homogeneity of agents in terms of their learning parameters, which simplified the mathematical description but could be relaxed and investigated further. Heterogeneity in agents' memory parameter $r$ was investigated in [28] where it was shown that a population containing both fast ($r = 1$) and slow ($r \ll 1$) agents still fragments across two markets, with the critical $\beta$ depending on the fraction of fast traders. Heterogeneity in $\beta$ might be mathematically more challenging but could in principle be tackled following the procedures outlined in [8]. The population can be split into subgroups of traders with the same $\beta$ whose steady-state market preference distributions should be found assuming fixed demand-to-supply market parameters. Finally, it should be checked whether those market aggregated parameters can be reproduced from the trader preferences obtained, i.e. whether the overall solution is self-consistent. This would be an interesting next step to investigate, together with heterogeneities in terms of trading strategies and budget constraints.

Data accessibility. Data available from the Dryad Digital Repository: https://doi.org/10.5061/dryad.cz8w9gj2n [29].

Competing interests. We declare we have no competing interests.

Authors' contributions. R.N., A.A. and P.S. conceptualized the model and formalized the mathematical framework. R.N. implemented the numerical simulations and prepared the figures. R.N., A.A. and P.S. analysed the results and discussed their implications. All authors wrote the manuscript and gave final approval for publication.

Funding. No funding has been received for this article.

Acknowledgements. The authors are grateful to Peter McBurney for useful discussions. P.S. acknowledges the stimulating research environment provided by the EPSRC Centre for Doctoral Training in Cross-Disciplinary Approaches to Non-Equilibrium Systems (CANES, EP/L015854/1). A.A. acknowledges the funding provided by the Institute of Physics Belgrade, through the grant by the Ministry of Education, Science, and Technological Development of the Republic of Serbia.

# Appendix A. Kramers–Moyal expansion

In this appendix, we give the expression of the drift and covariance matrix that appear in the Kramers–Moyal expansion in equation (3.1). We only give the results here; the steps of the derivation can be found in the thesis of Alorić [29]. First, the drifts of the attraction differences are

$$\mu_2^{(c)}(\boldsymbol{\Delta A}^{(c)}, f_1, f_2, f_3) = \left(\mathcal{P}_1^{(c)}(f_1)P(M=1) - \mathcal{P}_2^{(c)}(f_2)P(M=2)\right) - \Delta A_2^{(c)} \qquad (A\,1)$$

and

$$\mu_3^{(c)}(\boldsymbol{\Delta A}^{(c)}, f_1, f_2, f_3) = \left(\mathcal{P}_1^{(c)}(f_1)P(M=1) - \mathcal{P}_3^{(c)}(f_3)P(M=3)\right) - \Delta A_3^{(c)}. \qquad (A\,2)$$

Here $\mathcal{P}_m^{(c)}(f_m)$ is the average payoff of a trader from class $c$ at market $m$ and $P(M=m)$ is the probability to trade at market $m$, which depends on the vector $\Delta A^{(c)}$ of attraction differences. We do not write this dependence explicitly to lighten the notations. The $f_m$ are the market aggregates, i.e. buyer-to-seller ratios, at the three markets. In order to check the validity of our calculations we compared the dynamics of the aggregate $f_1$ during a multi-agent simulation with the evolution of the aggregates under the homogeneous population dynamics as detailed in [7], finding good agreement as shown in figure 7.

We next look at the covariance matrix of the effective noise acting on the attraction differences

$$\begin{pmatrix} \Sigma_{22}^{(c)} & \Sigma_{23}^{(c)} \\ \Sigma_{23}^{(c)} & \Sigma_{33}^{(c)} \end{pmatrix}, \qquad (A\,3)$$

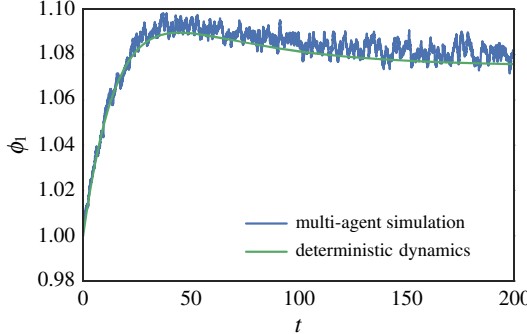

**Figure 7.** Comparison between the time series of the aggregate (ratio of buyers to sellers) at the first market during a multi-agent simulation (with $r = 0.01$ and $10^4$ agents in each class) and its evolution under the homogeneous population dynamics. The parameters for the plots in this figure are $(\theta_1, \theta_2, \theta_3) = (0.2, 0.5, 0.8)$, $\beta = 1/0.3$ and $p_{\mathcal{B}}^{(1)} = 1 - p_{\mathcal{B}}^{(2)} = 0.8$.

which is given by

$$
\Sigma_{22}^{(c)}(\mathbf{\Delta A}^{(c)}, f_1, f_2, f_3) = \left( \mathcal{Q}_1^{(c)}(f_1) - 2\Delta A_2^{(c)} \mathcal{P}_1^{(c)}(f_1) \right) P(M = 1)
$$
$$
+ \left( \mathcal{Q}_2^{(c)}(f_2) - 2\Delta A_2^{(c)} \mathcal{P}_2^{(c)}(f_2) \right) P(M = 2) + \Delta A_2^{(c)\,2}, \tag{A 4}
$$

$$
\Sigma_{33}^{(c)}(\mathbf{\Delta A}^{(c)}, f_1, f_2, f_3) = \left( \mathcal{Q}_1^{(c)}(f_1) - 2\Delta A_3^{(c)} \mathcal{P}_1^{(c)}(f_1) \right) P(M = 1)
$$
$$
+ \left( \mathcal{Q}_3^{(c)}(f_3) - 2\Delta A_3^{(c)} \mathcal{P}_3^{(c)}(f_3) \right) P(M = 3) \tag{A 5}
$$
$$
+ \Delta A_3^{(c)\,2}
$$

and

$$
\Sigma_{23}^{(c)}(\mathbf{\Delta A}^{(c)}, f_1, f_2, f_3) = \Delta A_2^{(c)} \left( P(M = 3) \mathcal{P}_3^{(c)}(f_3) - P(M = 1) \mathcal{P}_1^{(c)}(f_1) \right)
$$
$$
+ \Delta A_3^{(c)} \left( P(M = 2) \mathcal{P}_2^{(c)}(f_2) - P(M = 1) \mathcal{P}_1^{(c)}(f_1) \right) \tag{A 6}
$$
$$
+ P(M = 1) \mathcal{Q}_1^{(c)}(f_1) + \Delta A_2^{(c)} \Delta A_3^{(c)},
$$

where $\mathcal{Q}_m^{(c)}(f_m)$ is the average squared payoff, see [7].

# Appendix B. Freidlin–Wentzell theory

We describe in this section the large deviation methods we use to study multimodal attraction distributions in the steady state of our agents' learning dynamics. As explained in more detail in [7], steady-state attraction distributions for small $r$ will be peaked around the stable fixed points of the single agent dynamics. The shape of these peaks becomes Gaussian for $r \to 0$, with a covariance matrix proportional to $r$ that is straightforward to determine. Much more difficult to find are the *weights* of the peaks as these involve rare fluctuations of an agent making the transition from one peak to another. In one dimension, the problem is tractable as an explicit formula for the steady-state distribution of attractions can be given [6]. In higher dimensions detailed balance [31] would have a similar simplifying effect, but our single agent dynamics in the two-dimensional attraction space (for each class of agents) does not have this property.

In our approach, we therefore consider the peak weights in an attraction distribution as a result of the balance between transitions between the various peaks. We therefore need to find the rates for these transitions. To do this, note from the Kramers–Moyal expansion that the single agent learning is described by a Langevin equation with noise variance $O(r)$. For $r \to 0$, we are therefore looking for transition rates in a low noise limit. This allows us to use Freidlin–Wentzell theory, which deals with large deviations of Langevin dynamics in exactly this limit [32].

## B.1. Freidlin–Wentzell theory

We use Freidlin–Wentzell theory in the form developed in [33,34], which generalizes the Eyring–Kramers [35] formula for the rates of noise-activated transitions to non-conservative dynamics. We give a brief summary of those aspects of Freidlin–Wentzell theory that we use in our numerical application

and refer to [32] for a mathematically rigorous description and to [33] for a more statistical physics-oriented summary.

Freidlin–Wentzell theory is concerned with the transition rates between two stable states (here $A_1^\star$ and $A_2^\star$; below we drop the $\Delta$ from the notation for the attraction differences for brevity) of a non-conservative stochastic dynamics in the low noise limit. A general Langevin equation can be written in the form

$$\dot{A}(t) = \boldsymbol{\mu}(\mathbf{A}(t)) + \sqrt{r}[\Sigma(\mathbf{A}(t))]^{1/2}\boldsymbol{\xi}(t), \tag{B1}$$

where $\boldsymbol{\xi}(t)$ is white noise with unit covariance matrix. The drift $\boldsymbol{\mu}$ and the covariance matrix $\Sigma$ of the noise in the Langevin equation are given in [7] for our learning dynamics. In the generic version above, we have omitted the superscript $(c)$ indicating the class of agents we are considering, as well as the dependence of drift and noise covariance on the market aggregates.

Associated with the Langevin dynamics is an Onsager–Machlup action $\mathcal{S}[A]$ for any path $A(t)$

$$\mathcal{S}[A] = \int_{t_1}^{t_2} \frac{1}{2}\left(\dot{\mathbf{A}}(t) - \boldsymbol{\mu}(\mathbf{A}(t))\right)^T \Sigma^{-1}(\mathbf{A}(t))\left(\dot{\mathbf{A}}(t) - \boldsymbol{\mu}(\mathbf{A}(t))\right) dt. \tag{B2}$$

The action determines the probability of observing any path $[A(t)]$ according to

$$\Gamma_{1\to 2} \sim \exp\left(-\frac{\mathcal{S}[\mathbf{A}]}{r}\right), \tag{B3}$$

where $\sim$ means that the equality is true up to a prefactor (which depends on the time discretization used). The main Freidlin–Wentzell result we need is that the rate $\Gamma_{1\to 2}$ for a transition from $A_1^\star$ to $A_2^\star$ (*forward path*) is [32,36]

$$\Gamma_{1\to 2} \sim \exp\left(-\frac{\mathcal{S}_{1\to 2}^\star}{r}\right), \tag{B4}$$

where $\mathcal{S}_{1\to 2}^\star$ is the minimal action achievable by any path from $A_1^\star$ to $A_2^\star$ in the infinite time interval $(t_1, t_2) = (-\infty, \infty)$. The rate $\Gamma_{2\to 1}$ for the *reverse* transition from $A_2^\star$ to $A_1^\star$ is similarly $\Gamma_{2\to 1} \sim \exp(-\mathcal{S}_{2\to 1}^\star/r)$.

The attraction distributions we are after will consist of narrow (for small $r$) peaks at $A_1^\star$ and $A_2^\star$. The weights $\omega_1$ and $\omega_2$ of these two peaks, which represent the probability for an agent to be within each peak, must then be such that forward and backward transitions balance

$$\omega_1 \Gamma_{1\to 2} = \omega_2 \Gamma_{2\to 1} \tag{B5}$$

and

$$\frac{\omega_1}{\omega_2} \propto \exp\left(\frac{\mathcal{S}_{1\to 2}^\star - \mathcal{S}_{2\to 1}^\star}{r}\right). \tag{B6}$$

This expression shows that when the forward and backward minimal actions are not equal, then one of the two peaks will have an exponentially small weight as $r \to 0$. In practice, this is true when the action difference inside the exponential in (B 5) is large compared with $r$. If it is only of order $r$ or smaller, then we cannot say anything about the weights as we do not determine the prefactor in (B 5), though we would expect them to be of order unity.

## B.2. Finding the minimal action path numerically

Following the method of Bunin *et al.* [36], we find the minimal action by discretizing the path $[A(t)]$, evaluating the action as a function of this discretized path and then minimizing with respect to the (discretized) path. The path is discretized into 10 equally spaced time steps between $t = 0$ and $t = 10$; we found this choice of parameters to be a reasonable trade-off between the precision of our result and the complexity of minimizing the discretized action.

There are other methods for finding the minimal value of the action defined in equation (B 2), such as solving a Hamilton–Jacobi equation [33], but we chose to use the path discretization method because we found this to be more robust with respect to changes of model parameters. The discretization approach could also be improved further, using for example the geometric minimum action method [26], but we found that this was not necessary to achieve the desired precision. We tested this e.g. by benchmarking against closed-form results that can be obtained for $M = 2$ [6].

The numerical path optimization can be simplified by restricting attention to the *activation* part of the path. Generally, for a system with two stable fixed points $A_1^\star$ and $A_2^\star$ and one saddle point $\bar{A}$ between them, the optimal path starting from $A_1^\star$ will pass through the saddle point $\bar{A}$ and then relax to $A_2^\star$ following the relaxation dynamics $\dot{A}(t) = \boldsymbol{\mu}(A(t))$, equation (B 2) shows that the relaxation dynamics does not contribute to the total action as the integrand (the Lagrangian) vanishes identically along this section of the path. As a consequence, the problem of finding a minimal action path between $A_1^\star$ and $A_2^\star$ can be reduced to finding the minimal action path between $A_1^\star$ and $\bar{A}$, i.e. from the initial fixed point to the saddle. This restriction significantly improves the precision of the numerical path optimization.

With the above method, we can work out the action difference between any two fixed points of the single agent dynamics, as a function of the market aggregates. The values of these aggregates where the action difference between two single agent fixed points vanishes identify the points where the steady state attraction distribution of our learning can have more than one peak. Either side of these values, a single peak is dominant in the attraction distribution; which peak this is changes discontinuously at a zero action difference value of the market aggregates.

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
