## [Peer Review File · Royal Society Open Science]

Review History

RSOS-202233.R0 (Original submission)

Review form: Reviewer 1

Is the manuscript scientifically sound in its present form?

Yes

Are the interpretations and conclusions justified by the results?

Yes

Is the language acceptable?

Yes

Do you have any ethical concerns with this paper?

No

Have you any concerns about statistical analyses in this paper?

No

Recommendation?

Accept with minor revision (please list in comments)

Comments to the Author(s)

The paper is well written and addresses an interesting problem. I would appreciate some comments on the points raised below:

1. Have the coexistence of markets venues that has been observed as a long run outcome of the learning process (given certain parameters' configurations) an interpretation in terms of traders' bounded rationality? can the long run behavior be characterized in some way?
2. The parameter beta, representing the intensity of choice of agents, has been used as a "bifurcation" parameter to tune the degree of persistence of different venues. However, beta is a parameter that is the same for all traders. Can the authors justify such an assumption? Can they give some intuition on how outcomes are modified if a distributions of values of beta among traders is assumed?
3. How the model can help to understand empirical facts (that is the presence of multiple markets)?

Decision letter (RSOS-202233.R0)

Dear Dr Alorić

On behalf of the Editors, we are pleased to inform you that your Manuscript RSOS-202233 "Fragmentation in trader preferences among multiple markets: Market coexistence versus single market dominance" has been accepted for publication in Royal Society Open Science subject to minor revision in accordance with the referees' reports. Please find the referees' comments along with any feedback from the Editors below my signature.

Please submit your revised manuscript and required files (see below) no later than 7 days from today's (ie 05-Jul-2021) date. Note: the ScholarOne system will 'lock' if submission of the revision is attempted 7 or more days after the deadline. If you do not think you will be able to meet this deadline please contact the editorial office immediately.

on behalf of Professor Marta Kwiatkowska (Subject Editor)
openscience@royalsociety.org

Subject Editor Comments to Author:

We have received one referee report for your paper, which has identified several points of clarification, please address the comments carefully, highlighting the revisions in the manuscript and attaching a point by point rebuttal. We apologise, but due to the pandemic we were unable to secure additional referees.

Reviewer comments to Author:

Reviewer: 1

Comments to the Author(s)

The paper is well written and addresses an interesting problem. I would appreciate some comments on the points raised below:

1. Have the coexistence of markets venues that has been observed as a long run outcome of the learning process (given certain parameters' configurations) an interpretation in terms of traders' bounded rationality? can the long run behavior be characterized in some way?
2. The parameter beta, representing the intensity of choice of agents, has been used as a "bifurcation" parameter to tune the degree of persistence of different venues. However, beta is a parameter that is the same for all traders. Can the authors justify such an assumption? Can they give some intuition on how outcomes are modified if a distributions of values of beta among traders is assumed?
3. How the model can help to understand empirical facts (that is the presence of multiple markets)?

===PREPARING YOUR MANUSCRIPT===

Please ensure that you include an acknowledgements' section before your reference list/bibliography. This should acknowledge anyone who assisted with your work, but does not

qualify as an author per the guidelines at <https://royalsociety.org/journals/ethics-policies/openness/>.

===PREPARING YOUR REVISION IN SCHOLARONE===

-- Ensure that your data access statement meets the requirements at <https://royalsociety.org/journals/authors/author-guidelines/#data>. You should ensure that you cite the dataset in your reference list. If you have deposited data etc in the Dryad repository, please only include the 'For publication' link at this stage. You should remove the 'For review' link.

Author's Response to Decision Letter for (RSOS-202233.R0)

See Appendix A.

Decision letter (RSOS-202233.R1)

Dear Dr Alorić,

I am pleased to inform you that your manuscript entitled "Fragmentation in trader preferences among multiple markets: Market coexistence versus single market dominance" is now accepted for publication in Royal Society Open Science.

on behalf of Marta Kwiatkowska (Subject Editor)
openscience@royalsociety.org

Appendix A

Manuscript RSOS-202233

Response to Reviewers

Dear Professor Marta Kwiatkowska,

Thank you for giving us the opportunity to submit a revised draft of the manuscript “Fragmentation in trader preferences among multiple markets: Market coexistence versus single market dominance” for publication in the Royal Society Open Science. We appreciate the time and effort that you and the reviewer have dedicated to providing feedback on our manuscript and are grateful for the insightful comments on and valuable improvements to our paper. We have incorporated most of the suggestions made by the reviewers. Those changes are highlighted within the manuscript. Please see below, in violet, a point-by-point response to the reviewer’s comments and concerns. All page numbers refer to the revised manuscript file with tracked changes.

Reviewer comments to Author:

Reviewer: 1

The paper is well written and addresses an interesting problem.

Author response: Thank you!

1. Have the coexistence of markets venues that has been observed as a long run outcome of the learning process (given certain parameters' configurations) an interpretation in terms of traders' bounded rationality? can the long run behavior be characterized in some way?

Author response: Thank you for these questions and comments. The model we discuss in the paper can indeed be thought of as a population of boundedly rational traders as they do not act based on the globally shared information, nor do they optimise an underlying utility function. We have added a few sentences discussing this in the manuscript. Throughout the manuscript we discuss long run population behaviours (steady states) with particular interest in cases where those are multimodal distributions.

The revised text reads as follows on page 18:

“The proposed model contains an implicit assumption of bounded rationality as the agents do not optimise any utility function or aim to make the rational/optimal choice; instead their behaviour is based on their past and observed outcomes. Depending on the learning parameters the agents are tunable between trading randomly and a behaviour that repeats the most rewarding past choices. The agents do not possess knowledge about market mechanisms nor the existence of various different agents nor their scores, they only make decisions based on their past observations. In this regard, these assumptions violate rational agent assumptions due to the lack of information and lack of utility-optimising behaviour. Nonetheless, in the case of two markets it has been shown [7] that when the agents' memory is infinitely long ($\rightarrow 0$) and they do not update their preferences for options they did not try in the last steps, then the expected outcome under rational behaviour (Nash equilibrium) is retrieved.”

2. The parameter β , representing the intensity of choice of agents, has been used as a "bifurcation" parameter to tune the degree of persistence of different venues. However, β is a parameter that is the same for all traders. Can the authors justify such an assumption? Can they give some intuition on how outcomes are modified if a distributions of values of β among traders is assumed?

Author response: We thank the reviewer for this comment. Indeed, in the paper we make an assumption of agent homogeneity in terms of learning parameters. This is done for the sake of mathematical tractability and we recognise that it is a strong assumption. It is, however, in line with our goal to investigate if heterogeneity of traders' market preferences can emerge from initially homogeneous populations. In the revised manuscript we discuss the homogeneity of learning parameters together with some results on populations of traders with heterogeneous memory parameters, and how our method could be used to address a population with heterogeneous β parameters.

The revised text reads as follows [on pages 19 and 20]:

“To this end, we made an assumption of homogeneity of agents in terms of their learning parameters, which simplified the mathematical description but could be relaxed and investigated further. Heterogeneity in agents' memory parameter r was investigated in [28] where it was shown that a population containing both fast ($r=1$) and slow ($r \ll 1$) agents still fragments across two markets, with the critical β depending on the fraction of fast traders. Heterogeneity in β might be mathematically more challenging but could in principle be tackled following the procedures outlined in [8]. The population can be split into subgroups of traders with the same β whose steady state market preference distributions should be found assuming fixed demand-to-supply market parameters. Finally, it should be checked whether those market aggregated parameters can be reproduced from the trader preferences obtained, i.e. whether the overall solution is self-consistent. This would be an interesting next step to investigate, together with heterogeneities in terms of trading strategies and budget constraints.”

3. How the model can help to understand empirical facts (that is the presence of multiple markets)?

Author response: We thank the reviewer for raising this question and prompting us to clarify our stance. We understand that the multiple markets in reality might exist due to emergent trader loyalties (as proposed by our model) but also due to existing heterogeneities in both traders and markets. The model as it is offers a stylised representation of reality with the goal of investigating mechanisms that could lead to market coexistence or dominance, and as such at the moment it is not directly comparable with empirical findings, but rather provides insights and motivation for further experimental and mathematical exploration of agents adaptation and long term consequences of it.

The revised text reads as follows [on the page 18]:

“This research question is motivated by empirical observations of multiple markets coexisting and attracting loyal traders both in in-silico and real market competitions. Rather than aiming to reproduce market stylised facts, here we investigate mechanisms that might lead to a previously neglected phenomenon, namely, that multiple market

loyalties, and thus market coexistence, could emerge without any underlying heterogeneity of agents or markets and only as a consequence of the coadaptation of the agents.”